# Recombinant Production of a TRAF-Domain Lectin from Cauliflower: A Soluble Expression Strategy for Functional Protein Recovery in *E. coli*

**DOI:** 10.3390/ijms26178287

**Published:** 2025-08-26

**Authors:** Ana Káren de Mendonça Ludgero, Ana Luísa Aparecida da Silva, Luiz Henrique Cruz, Camila Aparecida Coelho Brazão, Kelly Maria Hurley Taylor, Leandro Licursi de Oliveira, Caio Roberto Soares Bragança, Christiane Eliza Motta Duarte

**Affiliations:** 1Laboratório de Sinalização Celular e Glicobiologia, Departamento de Ciências Biomédicas e Saúde, Universidade do Estado de Minas Gerais, Passos 37900-004, Brazil; 2Laboratório de Imunoquímica e Glicobiologia, Departamento de Biologia Geral, Universidade Federal de Viçosa, Viçosa 36570-900, Brazil; 3Laboratório de Fisiologia de Microrganismos, Departamento de Ciências Biomédicas e Saúde, Universidade do Estado de Minas Gerais, Passos 37900-004, Brazil

**Keywords:** *Brassica oleracea*, heterologous expression, optimization

## Abstract

Lectins are glycan-binding proteins involved in diverse biological processes and have gained attention for their potential applications in biotechnology and immunomodulation. BOL (*Brassica oleracea* lectin) is a unique ~34 kDa lectin isolated from *Brassica oleracea* var. *botrytis*, composed exclusively of TRAF-like domains, where TRAF stands for tumor necrosis factor receptor–associated factor. To overcome the limitations of plant-based extraction, we aimed to produce recombinant BOL in *Escherichia coli*. Various strains and expression vectors were tested under distinct induction conditions to optimize solubility and yield. While expression using pET28a was unsuccessful, GST-tagged BOL was efficiently expressed in *E. coli* BL21-R3-pRARE2(DE3) and purified using affinity chromatography. Functional assays demonstrated that the recombinant protein retained lectin activity, as evidenced by hemagglutination of goat erythrocytes. Protein identity was confirmed by MALDI-TOF/TOF mass spectrometry, with tryptic peptides matching the BOL lectin sequence in the National Center for Biotechnology Information (NCBI) database. Our findings highlight the importance of codon optimization, temperature modulation, and fusion tag selection for the successful expression of eukaryotic lectins in *E. coli*. This work provides a platform for future functional studies of BOL and supports its potential application in plant immunity and biomedical research.

## 1. Introduction

Lectins are a diverse family of carbohydrate-binding proteins, each characterized by one or more non-catalytic carbohydrate-recognition domains (CRDs) that bind specific glycans reversibly and in a non-covalent manner, without inducing biochemical modification. Through multivalent and reversible binding of their CRDs to glycan moieties on the surfaces of erythrocytes or soluble glycoproteins, lectins can cross-link glycosylated structures. These interactions enable lectins to agglutinate erythrocytes and precipitate glycoconjugates [1]. Found ubiquitously across all domains of life—including viruses, bacteria, fungi, animals, and plants—lectins mediate a wide array of biological functions. These include pathogen attachment to host cells [2,3], defense against insect herbivory and microbial parasites [4], and recognition of symbiotic partners through glycan-specific interactions [5]. In each case, the lectin–glycan interaction functions as a molecular recognition event that initiates or stabilizes biological processes such as cellular adhesion, immune signaling, or microbial colonization, often through multivalent binding to densely glycosylated surfaces.

Beyond their native biological roles, lectins have been exploited in biotechnology and biomedical research due to their mitogenic, antiproliferative, antimicrobial, and immunomodulatory properties [6,7]. However, large-scale purification from natural sources presents several limitations. Chromatographic techniques, although commonly used, frequently fail to separate structurally similar isoforms and typically result in mixtures that impair functional and structural characterization. Moreover, they usually provide very low yield [8].

Among plant-derived lectins, BOL (*Brassica oleracea* lectin)—a ~34 kDa protein isolated from *Brassica oleracea* var. *botrytis* (cauliflower)—is unique. It comprises TRAF-like domains, formally known as the TNF-receptor associated factor (TRAF) domain, also referred to as the meprin and TRAF-C homology (MATH) domain. In other biological contexts, this domain is primarily associated with protein–protein interactions and is not traditionally linked to glycan binding. However, BOL has been shown to bind complex N-linked oligosaccharides present in glycoproteins such as fetuin and asialofetuin. This finding reveals a novel glycan-recognition mechanism within a protein composed exclusively of TRAF-like domains [9,10]. Preliminary biochemical analyses indicate that BOL is thermostable between 4 °C and 60 °C, exhibits optimal activity at pH 7.0–8.0, and induces macrophage activation and nitric oxide production—suggesting a potential role as an immunostimulatory agent. Three chromatographic steps are required to obtain purified preparation from *Brassica oleracea* var. *botrytis* [9].

In contrast, recombinant expression offers a scalable and controllable alternative, circumventing the limitations of plant-based extraction, improving yield, and simplifying purification protocols [8,11]. *Escherichia coli* remains the host of choice for recombinant lectin production due to its rapid growth rate, genetic tractability, and availability of diverse expression vectors [12,13]. Nonetheless, overexpression of lectins in *E. coli* often results in the formation of insoluble inclusion bodies, protein misfolding, and degradation by host proteases [11,14]. Strategies such as reducing induction temperature (12–20 °C), extending expression duration, and fine-tuning inducer concentration can mitigate these issues by promoting proper folding and reducing proteolysis [15,16,17].

Fusion tags enhance both solubility and purification. The polyhistidine tag (His_6_) enables straightforward purification via immobilized metal affinity chromatography (IMAC) under native conditions [18], whereas the glutathione S-transferase (GST) tag improves solubility and allows efficient purification through glutathione affinity chromatography [19,20].

Our group previously identified BOL as a lectin harboring two MATH domains, structurally related to the TRAF protein family [9,10]. In this study, we report the heterologous expression of BOL in *E. coli*. While each MATH domain typically comprises seven to eight antiparallel β-strands mediating protein–protein interactions in animal TRAFs [21], their function in plant proteins remains poorly understood. In *Arabidopsis thaliana*, TRAF/MATH-only proteins have been associated with autophagy regulation, innate immunity, and gametophyte development [22]. BOL represents the first reported lectin composed exclusively of TRAF/MATH-like domains with glycan-binding capacity.

Here, we describe the systematic optimization of BOL expression in multiple *E*. *coli* strains, including BL21(DE3), C41(DE3), ArcticExpress(DE3), and BL21-R3-pRARE2. We tested two fusion tags and various induction parameters to maximize soluble expression. Our goal is to establish a reliable platform for producing functional BOL, enabling downstream functional assays, subcellular localization studies in planta, and exploration of its biotechnological potential in plant defense and immune modulation.

## 2. Results

The primary objective of this study was to optimize the heterologous expression of the BOL lectin in *Escherichia coli*. To this end, various *E. coli* strains and two expression vectors were tested under a range of culture conditions, including temperature and isopropyl β-D-1-thiogalactopyranoside (IPTG) concentration, to identify the best combination for high-yield, soluble expression. This experimental strategy was guided by previous expression attempts using the pET21b vector, in which low expression levels and poor solubility were observed across two different host strains and induction conditions [10]. These initial findings prompted the evaluation of alternative vectors: pET28a, containing an N-terminal His-tag, and pGEX-4T-1, with an N-terminal GST tag, combined with optimized induction protocols in multiple *E. coli* strains.

### 2.1. Evaluation of BOL Expression in E. coli BL21(DE3) and C41(DE3) Using pET28a

Initial expression trials were performed using the pET28a vector encoding a 6×His-tagged BOL construct. In *E. coli* BL21(DE3), induction at 37 °C for 4 h with IPTG at 0.5, 1.0, and 1.5 mM resulted in no visible bands corresponding to the expected molecular weight (~34 kDa), as shown by SDS-PAGE (Figure 1A). Similarly, in *E. coli* C41(DE3), IPTG induction at 0.5–3.0 mM failed to produce detectable expression (Figure 1B). These results suggest that the pET28a system is unsuitable for the expression of BOL under the tested conditions.

### 2.2. Expression of BOL-GST in E. coli BL21(DE3)

Due to the lack of expression with pET28a, the BOL gene was subcloned into the pGEX-4T-1 vector. In BL21(DE3), expression was induced at 37 °C for 4 h with 0.5–1.5 mM IPTG. SDS-PAGE analysis revealed consistent expression across IPTG concentrations, but the majority of the recombinant BOL-GST (~60 kDa) accumulated in the insoluble fraction, indicating predominant formation of inclusion bodies (Figure 2).

### 2.3. Expression of BOL-GST in E. coli C41(DE3)

To improve solubility, strain C41(DE3) was tested using the pGEX-4T-1 vector under varying IPTG concentrations (0.05 to 1 mM) at 37 °C (Figure 3A). Although this strain exhibited higher apparent expression levels compared to BL21(DE3), the recombinant protein predominantly accumulated in the insoluble fraction. Western blot analysis confirmed the expression of BOL-GST in whole-cell lysates (Figure 3B).

Subsequently, induction at 20 °C overnight with 0.1 mM IPTG was attempted to reduce aggregation. This condition modestly improved solubility, although overall yield decreased (Figure 3C).

### 2.4. Evaluation of ArcticExpress(DE3) for Low-Temperature Expression

Given the persistence of inclusion bodies, ArcticExpress(DE3)—which co-expresses cold-adapted chaperonins—was evaluated. Cultures were induced at 12 °C for 24 h with 0.01–1.0 mM IPTG (Figure 4A). However, SDS-PAGE revealed overlapping bands between BOL-GST and endogenous chaperonins, even in uninduced and untransformed samples, compromising band resolution. Western blotting confirmed nonspecific signals, suggesting that ArcticExpress(DE3) is not suitable for BOL-GST expression under these conditions (Figure 4B).

### 2.5. High-Yield Expression of BOL-GST in E. coli BL21-R3-pRARE2

The BL21-R3-pRARE2(DE3) strain, which co-expresses rare tRNAs, was tested for improved expression. Expression assays were performed using 0.1 mM IPTG at both 37 °C and 20 °C (Figure 5A,B). Although SDS-PAGE analyses revealed comparable expression levels of the BOL-GST fusion protein at both temperatures, a higher yield of purified protein was obtained from cultures induced at 37 °C. Despite partial formation of inclusion bodies, this condition allowed successful purification of soluble protein using GST-affinity chromatography, yielding approximately 6.5 mg/100 mL culture. Western blot analysis confirmed the presence of BOL-GST in total, soluble, and purified fractions (Figure 5C).

Densitometric analysis of SDS-PAGE gels was performed to estimate the relative distribution of BOL-GST between the soluble and insoluble fractions under all tested conditions. Band intensities were normalized to the total protein signal in each lane, enabling a comparative assessment of BOL-GST solubility across the four *E. coli* strains: BL21(DE3), C41(DE3), ArcticExpress(DE3), and BL21-R3-pRARE2(DE3). The results are expressed as percentages of total expression and are shown in Figure 6.

### 2.6. Functional Activity and Identity Confirmation of Recombinant BOL-GST

The hemagglutination assay demonstrated that the purified recombinant BOL-GST retains carbohydrate-binding activity, effectively agglutinating goat erythrocytes (Figure 7A). The hemagglutination titer of recombinant BOL-GST was 1:8, indicating that the fusion protein preserves its native functional properties.

Mass spectrometry analysis (MALDI-TOF/TOF) of the purified fusion protein verified its identity. Trypsin-digested peptides matched the sequence of *Brassica oleracea* BOL lectin in the NCBI database (MN937259), confirming the correct expression and purification of the target protein (Figure 7B).

## 3. Discussion

Although *E. coli* BL21(DE3) is one of the most widely used hosts for heterologous protein expression, its performance is highly dependent on the nature of the target protein. In the present study, no detectable expression of BOL-6xHis was observed in BL21(DE3). *E. coli* lacks the eukaryotic chaperones and post-translational machinery required for the proper folding of complex proteins like BOL, increasing the risk of misfolding, aggregation, or degradation. In some instances, the recombinant protein may exert cytotoxic effects, impairing host viability and thus reducing overall yield [23].

Despite the lack of detectable expression of BOL-6xHis in BL21(DE3), this has proven effective for expressing other recombinant proteins. For example, ref. [24] reported successful expression of recombinant Cas9 in both BL21(DE3) and BL21(DE3) Rosetta strains. Cas9 was readily detected in crude lysates and efficiently purified using Ni-NTA affinity chromatography. Notably, expression levels were lower in the Rosetta strain, and detection required Western blot analysis post-purification.

For BOL-6xHis, alternative expression attempts in *E. coli* C41(DE3) also failed to yield detectable protein by SDS-PAGE. The pET series vectors, including pET28a used in this study, rely on the T7 promoter, a bacteriophage-derived regulatory element located immediately upstream of the multiple cloning site (MCS), to drive high-level transcription of the target gene in host strains that express T7 RNA polymerase. Despite its efficiency, the T7-based system can impose a considerable metabolic burden on the host cells, often leading to reduced viability, impaired folding capacity, or accumulation of inclusion bodies—all of which can compromise recombinant protein yield [16].

In contrast, in a previous study from our group, detectable expression of BOL-6xHis was achieved using the pET21b vector in BL21(DE3) cells, although the recombinant protein accumulated predominantly in the insoluble fraction [10]. While both vectors are driven by the same T7 promoter, they differ in important architectural features, including the position of the affinity tag, backbone elements, and untranslated regions that may influence transcriptional or translational efficiency. Despite the lack of a direct comparison under matched conditions in this study, these differences may partially explain the divergent expression outcomes observed. In light of these challenges, we turned to the BOL-GST construct as an alternative strategy. Preliminary trials using BL21(DE3) and C41(DE3) at different IPTG concentrations revealed that C41(DE3) provided a more favorable expression profile. C41(DE3) and C43(DE3), often referred to as “Walker strains,” are engineered derivatives of BL21(DE3) designed to enhance expression of toxic or difficult-to-fold proteins. Specifically, C41(DE3) harbors mutations that alleviate protein-induced cytotoxicity and reduce inclusion body formation in certain contexts [23].

Saribas et al. (2001) [25] demonstrated the superior performance of C41(DE3) in expressing membrane-associated cytochrome P450 2B4 at both small and large scales. Compared to JM109(DE3)pLysS, which suffered from plasmid instability, poor induction, and toxicity, C41(DE3) ensured stable plasmid maintenance and consistent expression. In our case, however, a large proportion of BOL-GST still accumulated in inclusion bodies. To address this, optimization steps targeting IPTG concentration, induction time, and temperature were implemented. While lower IPTG concentrations typically reduce total yield, they favor soluble protein expression by minimizing metabolic stress and inclusion body formation. Conversely, higher IPTG levels can exacerbate cellular stress and hinder protein production [16]. Balancing induction strength with host viability is critical when expressing eukaryotic proteins in *E. coli*.

Temperature modulation is another key factor influencing recombinant protein solubility and activity. Huang et al. (2021) [26] demonstrated that expression of the HCV NS3 antigen at 20 °C, instead of 37 °C, improved folding efficiency, reduced aggregation, and enhanced enzymatic activity. This improvement was attributed to slower translation kinetics and more effective co-translational folding at lower temperatures.

In line with this, the ArcticExpress(DE3) strain—engineered to co-express cold-adapted chaperonins Cpn10 and Cpn60 from *Oleispira antarctica*—allows induction at 10–13 °C (Agilent Technologies, Inc., Santa Clara, CA, USA, 2015). We tested this strain for BOL-GST expression, but SDS-PAGE analysis revealed a prominent 60 kDa band in both non-induced and non-transformed controls, suggesting the signal overlapped with endogenous chaperonin bands.

Similar challenges were reported by Guo et al. (2022) [27], who expressed murine cytosolic carboxypeptidase 6 (CCP6) in ArcticExpress(DE3). Although CCP6 was soluble, it co-purified with Cpn60, resulting in inactive protein. Treating lysates with ATP, Mg^2+^, and K^+^ promoted chaperonin dissociation and restored enzymatic activity. Annamalai et al. (2009) [28] also observed chaperonin contamination when expressing *Mycobacterium tuberculosis* topoisomerase I in this strain. However, due to the clear molecular weight difference (~102.3 kDa vs. 55–66 kDa), identification was straightforward. In our case, the BOL-GST fusion protein (~60 kDa) co-migrated with Cpn60, making interpretation and purification unreliable. Thus, ArcticExpress(DE3) was deemed unsuitable.

To further enhance soluble yield, we employed *E. coli* BL21-R3-pRARE2(DE3), a strain supplemented with tRNAs for seven rare codons that are poorly represented in *E. coli* but frequently used in eukaryotic genes [29,30]. Rare codon bias can impair translation efficiency and folding of heterologous proteins. Using this strain, we successfully expressed BOL-GST in soluble form.

A similar approach was reported by Reiss et al. (2014) [31] for expression of the *choA* gene from *Chryseobacterium gleum*, which contains ~8% rare codons. Although expression in a JM109 strain co-transformed with pRARE2 was successful, the protein aggregated unless induced at 16 °C. Burgess-Brown et al. (2008) [32] also demonstrated that rare codon-supplemented strains improved both yield and solubility of eukaryotic proteins. Codon optimization and rare tRNA supplementation were shown to be complementary strategies, enhancing not only expression but also purification efficiency.

In contrast to these findings, which emphasize low-temperature induction, we found that BOL-GST expression levels in the BL21-R3-pRARE2(DE3) strain were comparable at 20 °C and 37 °C, as indicated by SDS-PAGE analysis (Figure 5A,B). Notably, induction at 37 °C resulted in a higher yield of soluble and purifiable protein. Although lower temperatures are generally used to enhance solubility by reducing aggregation and promoting proper folding, this pattern is not universal. In our case, the elevated expression rate at 37 °C appears to have offset the partial formation of inclusion bodies, ultimately leading to greater recovery of functional protein. Similar findings have been reported, in which high expression levels at 37 °C were sufficient to compensate for reduced solubility, thereby improving overall protein yield [33,34].

Our results validate the optimized conditions employed in this study for the heterologous expression of BOL-GST. The recombinant protein obtained from BL21-R3-pRARE2(DE3) lysates was soluble, retained lectin activity—as demonstrated by hemagglutination assays—and was confirmed by mass spectrometry. These findings underscore the value of combining low-temperature induction, rare codon supplementation, and alternative fusion tags to overcome the challenges of expressing eukaryotic proteins in *E. coli*. Ultimately, the successful production of soluble and active BOL-GST underscores the effectiveness of the combined strategy and lays the groundwork for exploring its functional roles in glycan recognition, immunomodulation, and potential applications in plant defense and biotechnology.

## 4. Materials and Methods

### 4.1. Plasmid Synthesis and Propagation

The BOL lectin coding sequence was chemically synthesized by GenOne Soluções em Biotecnologia (Rio de Janeiro, Brazil). The synthetic gene was cloned into the expression vectors pGEX-4T-1 (Amersham Biosciences^®^, San Diego, CA, USA) and pET28a (Novagen^®^, Madison, WI, USA). For cloning purposes, an *Eco*RI restriction site (GAATTC) was added at the 5′ end, and a *Sal*I site (GTCGAC) at the 3′ end of the sequence. These sites enabled directional cloning into the respective vectors, yielding the constructs pGEX-4T-1_BOL and pET28a_BOL.

The resulting plasmids were propagated in *Escherichia coli* DH5α and isolated using the Fast-n-Easy Plasmid MiniPrep Kit (Cellco^®^, São Paulo, Brazil), following the manufacturer’s instructions. Purified plasmid DNA was subsequently used to transform the *E. coli* expression strains BL21(DE3), C41(DE3), ArcticExpress(DE3), and BL21-R3-pRARE2.

### 4.2. Transformation of Escherichia coli by Heat Shock

Chemically competent cells of *E. coli* strains BL21(DE3), C41(DE3), ArcticExpress(DE3), and BL21-R3-pRARE2 were transformed with 150 ng of plasmid DNA using the standard heat shock protocol. DNA was gently mixed with 50 µL of competent cells and incubated on ice for 10 min, followed by heat shock at 42 °C for 45 s and cooling on ice for 2 min. This cycle was repeated twice to enhance transformation efficiency.

Following heat shock, 800 µL of Luria–Bertani (LB) broth was added, and cells were incubated at 37 °C for 1 h with agitation (300 rpm). After centrifugation (12,000× *g* rpm, 1 min, Hettich MIKRO 220R, Hettich GmbH & Co., Tuttlingen, Germany), cell pellets were resuspended and plated on LB agar containing the appropriate antibiotics: kanamycin (100 mg/L) for pET28a, ampicillin (100 mg/L) for pGEX-4T-1, gentamicin (20 mg/L) for ArcticExpress(DE3), and chloramphenicol (25 mg/L) for BL21-R3-pRARE2.

### 4.3. Recombinant Expression of BOL Lectin

#### 4.3.1. Expression Using pET28a Vector

Initial expression trials were conducted in *E. coli* BL21(DE3) and C41(DE3). A 500 µL aliquot of overnight-grown pre-inoculum was transferred into 5 mL of fresh LB medium supplemented with kanamycin (100 mg/L) and incubated overnight at 37 °C with shaking at 250 rpm (MaxQ™ 4000 shaker, ThermoFisher^®^, Marietta, OH, USA). Induction was carried out by adding isopropyl β-D-1-thiogalactopyranoside (IPTG) at final concentrations of 0.5, 1.0, or 1.5 mM, followed by 4 h incubation at 37 °C with shaking.

#### 4.3.2. Expression Using pGEX-4T-1 Vector

The pGEX-4T-1 construct was expressed in *E. coli* BL21(DE3), C41(DE3), ArcticExpress(DE3), and BL21-R3-pRARE2. Small-scale expression assays were conducted in 5 mL of LB medium supplemented with ampicillin (100 mg/L), except for ArcticExpress(DE3), which required no additional antibiotics. A 500 µL aliquot of pre-inoculum was transferred into 5 mL of fresh LB and incubated at 37 °C with shaking at 250 rpm. Protein expression was induced with IPTG (0.5, 1.0, or 1.5 mM), followed by 4 h incubation at 37 °C.

For ArcticExpress(DE3), an additional strategy was tested: 100 µL of pre-inoculum was incubated at 30 °C for 3 h, cooled to 12 °C for 10 min, and induced with IPTG (0.05–1.0 mM). Based on initial results, subsequent assays used 0.1 mM IPTG and were incubated for 24 h at 12 °C.

Large-scale expression was performed in *E. coli* BL21-R3-pRARE2. A 2 mL aliquot of pre-inoculum was transferred into 200 mL of LB medium with ampicillin (100 mg/L) and grown in 1 L Erlenmeyer flasks at 37 °C with agitation (220 rpm) for 8 h. Induction was performed with 0.1 mM IPTG, followed by overnight incubation at 20 °C with agitation.

### 4.4. Sample Collection and Analysis

Post-induction, 600 µL samples were centrifuged (10,000× *g* rpm, 5 min, 4 °C Hettich MIKRO 220R, Hettich GmbH & Co., Tuttlingen, Germany). Supernatants were discarded and pellets stored at −20 °C. Additional aliquots were lysed in 250 µL of buffer (20 mM Tris-HCl, 100 mM NaCl, 1% Triton X-100, 1 mM Phenylmethylsulfonyl fluoride-PMSF) using a Sonics Vibra Cell VCX 130 sonicator (Sonics & Materials, Inc., Newtown, CT, USA) (3 mm probe; 4 × 20 s pulses at 30% amplitude). Lysates were centrifuged, and soluble (supernatant) and insoluble (pellet) fractions were analyzed by SDS-PAGE (12% gels).

### 4.5. SDS-PAGE and Densitometry Analysis

Electrophoresis was performed under denaturing conditions using a Mini-PROTEAN Tetra Vertical Electrophoresis Cell system (Bio-Rad^®^, Bedok, Singapore). Gels consisted of stacking and resolving layers (12%). Electrophoresis was conducted at 80 V for 15 min, followed by 150 V until separation was complete. Gels were stained with Coomassie Brilliant Blue G-250 for 6 min and destained with 30% methanol and 10% acetic acid. Images were acquired with a ChemiDoc MP system (Bio-Rad^®^, Bedok, Singapore).

Gel images were acquired with a ChemiDoc MP system and analyzed using ImageJ software (v. 2.1.0/1.53 c, National Institutes of Health, Bethesda, MD, USA—NIH). To enable comparative assessment between samples, for each lane, the integrated density of the BOL band was background-corrected and normalized to the total integrated density of all protein bands in the same lane. Given that samples were loaded with different total protein amounts, these densitometric values are interpreted as semi-quantitative estimates of the relative distribution of BOL between soluble and insoluble fractions rather than absolute yield measurements. All raw images and the numerical spreadsheets used for calculations are provided in the Appendix A.

### 4.6. Western Blot

Proteins were transferred to nitrocellulose membranes using the Mini Trans-Blot^®^ wet transfer system (Bio-Rad^®^ Bedok, Singapore) in buffer (50 mM Tris, 40 mM glycine, 20% methanol) at 350 mA for 1 h. Membranes were washed in TBST (3×), blocked with PBS-casein (Thermo Scientific^®^, Rockford, IL, USA) for 1 h, and incubated overnight with mouse anti-GST primary antibody (cat. no. MA4-004, clone 8-326, Invitrogen^®^, 1:1000, Burlington, ON, Canada). After washing, membranes were incubated for 1 h with HRP-conjugated goat anti-mouse IgG (Invitrogen^®^, cat. no. 621040, 1:10,000, Burlington, ON, Canada). Detection was performed by chemiluminescence (SuperSignal™ West Dura, Thermo Scientific^®^, Rockford, IL, USA) and captured with a ChemiDoc MP system (Bio-Rad^®^, Bedok, Singapore).

### 4.7. Purification of the BOL-GST Fusion Protein

BOL-GST was purified using the Pierce™ GST Spin Purification Kit (Thermo Scientific™, Rockford, IL, USA). Columns were equilibrated with binding/wash buffer (125 mM Tris, 150 mM NaCl, pH 8.0). Protein extracts diluted in this buffer were loaded onto the column and incubated at 4 °C for 1 h on a rocking platform. After centrifugation (700× *g*, 2 min), unbound fractions were collected. Columns were washed (3×), and bound proteins eluted in three steps using 10 mM reduced glutathione. Eluted fractions were analyzed by SDS-PAGE, and protein concentration was determined by absorbance at 280 nm (Multiskan SkyHigh, Thermo Fisher^®^, Singapore).

### 4.8. Mass Spectrometry and Protein Identification

The protein band corresponding to BOL was excised from the SDS-PAGE gel and destained with 50% acetonitrile and 25 mM ammonium bicarbonate. Samples were reduced (65 mM DTT, 30 min, 56 °C), alkylated (200 mM iodoacetamide, 30 min, RT), dehydrated, and concentrated (SpeedVac™, Waltham, MA, USA). In-gel digestion was performed overnight at 37 °C using 2.5 µg/mL trypsin in 10% acetonitrile and 40 mM ammonium bicarbonate (pH 8.0). Peptides were extracted (50% acetonitrile, 5% formic acid), dried, and resuspended in 0.1% formic acid. Desalting was performed using ZipTip C18 (Merck KGaA, Darmstadt, Germany). MALDI-TOF/TOF analysis was carried out with an Ultraflex III spectrometer (Bruker Daltonics, Bremen, Germany) in reflector mode (*m*/*z* 640–3240). Protein identification was performed using Mascot (in-house server) against the NCBI database.

### 4.9. Hemagglutination Activity Assay

Hemagglutination activity was assessed by serial two-fold dilutions (50 µL) of recombinant BOL in U-bottom microtiter plates, followed by the addition of 25 µL of a 2% suspension of goat erythrocytes. The hemagglutination titer was defined as the reciprocal of the highest dilution showing visible agglutination.

## Figures and Tables

**Figure 1 ijms-26-08287-f001:**
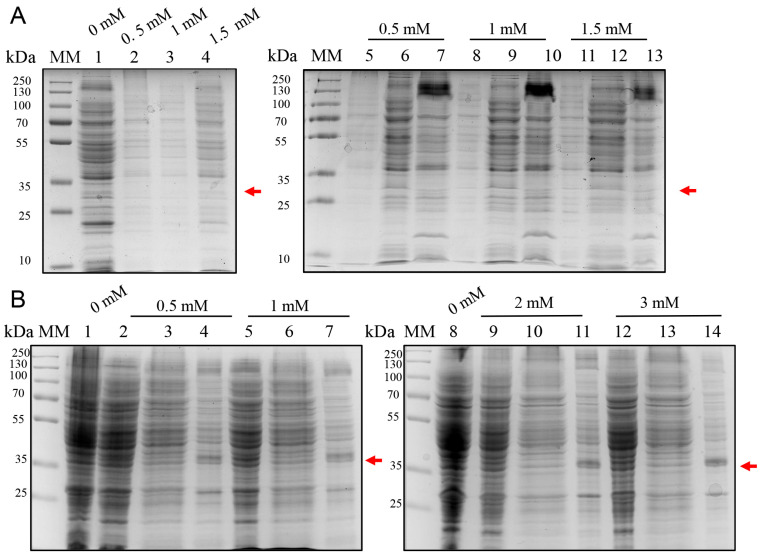
SDS-PAGE analysis of BOL-6xHis expression in *E. coli* BL21(DE3) and C41(DE3) transformed with the pET28a vector. Cultures were grown at 37 °C for 4 h and induced with varying IPTG concentrations. Cells were lysed by sonication and clarified by centrifugation to separate soluble and insoluble protein fractions. The expected molecular weight of the BOL-6xHis fusion protein (~34 kDa) is indicated by a red arrow. MM: molecular weight marker (PageRuler™ Prestained Protein Ladder, Thermo Scientific™, Vilnius, Lithuania). Gels were stained with Coomassie Brilliant Blue G-250. (**A**) Expression in *E. coli* BL21(DE3) induced with 0.5, 1.0, and 1.5 mM IPTG. Lane 1: non-induced control (total cell lysate); Lanes 2–4: total cell lysates of cultures induced with 0.5, 1.0, and 1.5 mM IPTG, respectively; Lanes 5–7: samples from cultures induced with 0.5 mM IPTG—total cell lysate (5), soluble protein fraction (6), insoluble protein fraction (7); Lanes 8–10: 1.0 mM IPTG—total cell lysate (8), soluble fraction (9), insoluble fraction (10); Lanes 11–13: 1.5 mM IPTG—total cell lysate (11), soluble fraction (12), insoluble fraction (13). (**B**) Expression in *E. coli* C41(DE3) induced with 0.5, 1.0, 2.0, and 3.0 mM IPTG. Lanes 1 and 8: non-induced control (total cell lysate); Lanes 2–4: 0.5 mM IPTG—total cell lysate (2), soluble fraction (3), insoluble fraction (4); Lanes 5–7: 1.0 mM IPTG—total cell lysate (5), soluble fraction (6), insoluble fraction (7); Lanes 9–11: 2.0 mM IPTG—total cell lysate (9), soluble fraction (10), insoluble fraction (11); Lanes 12–14: 3.0 mM IPTG—total cell lysate (12), soluble fraction (13), insoluble fraction (14). No detectable bands corresponding to the expected size of the BOL-6xHis fusion protein were observed under any of the tested conditions.

**Figure 2 ijms-26-08287-f002:**
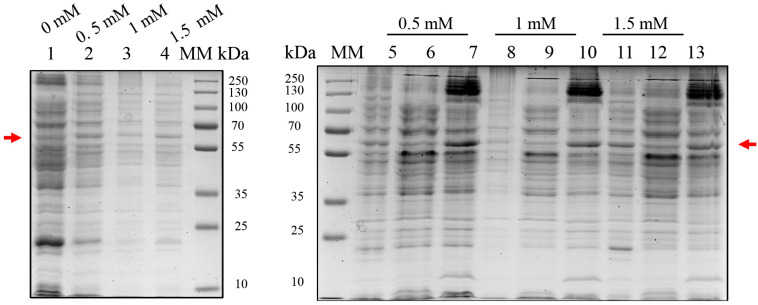
SDS-PAGE analysis of BOL-GST expression in *E. coli* BL21(DE3) using the pGEX-4T-1 vector. Cultures were grown at 37 °C for 4 h and induced with different IPTG concentrations (0.5, 1.0, and 1.5 mM). Cells were lysed by sonication and clarified by centrifugation to separate soluble and insoluble protein fractions. The expected molecular weight of the BOL-GST fusion protein (~60 kDa) is indicated by a red arrow. MM: molecular weight marker (PageRuler™ Prestained Protein Ladder, Thermo Scientific™, Vilnius, Lithuania). Gels were stained with Coomassie Brilliant Blue G-250. Lane 1: non-induced control (total cell lysate); Lanes 2–4: total cell lysates of cultures induced with 0.5, 1.0, and 1.5 mM IPTG, respectively; Lanes 5–7: 0.5 mM IPTG—total cell lysate (5), soluble protein fraction (6), insoluble protein fraction (7); Lanes 8–10: 1.0 mM IPTG—total cell lysate (8), soluble fraction (9), insoluble fraction (10); Lanes 11–13: 1.5 mM IPTG—total cell lysate (11), soluble fraction (12), insoluble fraction (13). Under all conditions tested, a high proportion of the BOL-GST fusion protein was consistently detected in the insoluble fractions.

**Figure 3 ijms-26-08287-f003:**
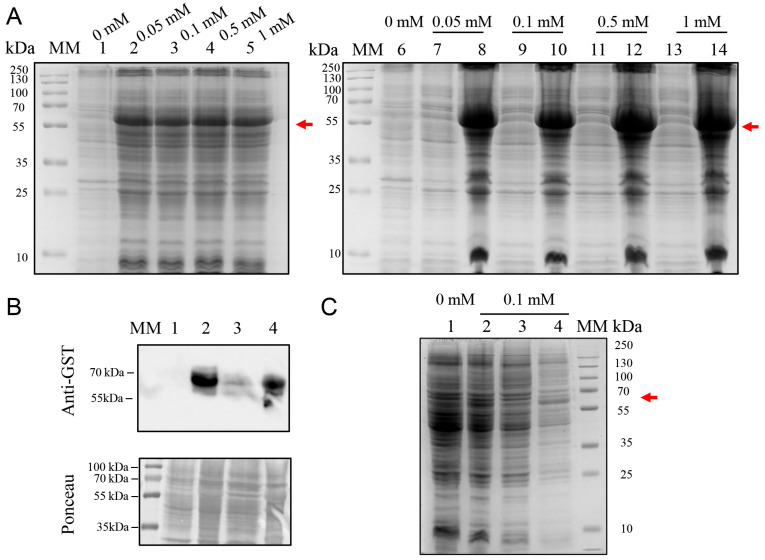
SDS-PAGE and Western blot analysis of BOL-GST expression in *E. coli* C41(DE3) using the pGEX-4T-1 vector under various induction conditions. Cultures were induced at 37 °C or 20 °C with different IPTG concentrations. Cells were lysed by sonication and clarified by centrifugation to separate soluble and insoluble protein fractions. The expected molecular weight of the BOL-GST fusion protein (~60 kDa) is indicated by a red arrow. MM: molecular weight marker (PageRuler™ Prestained Protein Ladder, Thermo Scientific™, Vilnius, Lithuania). Gels were stained with Coomassie Brilliant Blue G-250. (**A**) SDS-PAGE analysis of cultures induced at 37 °C for 4 h with IPTG concentrations of 0.05, 0.1, 0.5, and 1.0 mM. Lane 1 and 6: non-induced control (total cell lysate); Lanes 2–5: total cell lysates induced with 0.05, 0.1, 0.5, and 1.0 mM IPTG, respectively; Lanes 7–8: 0.05 mM IPTG—soluble fractions (7), insoluble fraction (8); Lanes 9–10: 0.1 mM IPTG—soluble fractions (9), insoluble fraction (10); Lanes 11–12: 0.5 mM IPTG—soluble fractions (11), insoluble fraction (12); Lanes 13–14: 1.0 mM IPTG—soluble fractions (13), insoluble fraction (14). Most recombinant protein was detected in the insoluble fractions, with similar expression levels across conditions and increased aggregation at 0.5 and 1.0 mM IPTG. (**B**) Western blot of total cell lysates confirming BOL-GST expression at 0.1 mM IPTG. Lane 1: non-induced control (total cell lysate); Lane 2: total cell lysate post-induction at 1.0 mM IPTG; Lane 3: soluble fraction; Lane 4: insoluble fraction. (**C**) SDS-PAGE analysis of cultures induced overnight at 20 °C with 0.1 mM IPTG. Lane 1: non-induced control (total cell lysate); Lane 2: total cell lysate post-induction; Lane 3: soluble fraction; Lane 4: insoluble fraction. Lower levels of insoluble protein accumulation were observed at 20 °C, albeit with reduced overall expression.

**Figure 4 ijms-26-08287-f004:**
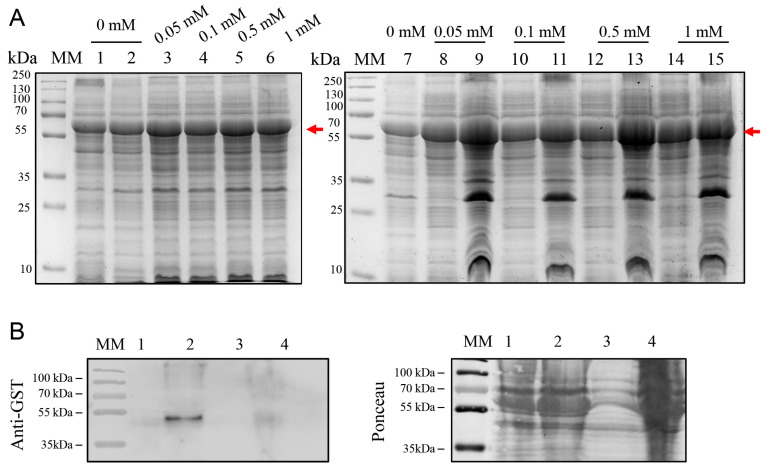
SDS-PAGE and Western blot analysis of BOL-GST expression in *E. coli* ArcticExpress (DE3) transformed with the pGEX-4T-1 vector. Cultures were grown at 37 °C for 3 h and induced at 12 °C for 24 h with different IPTG concentrations (0.05, 0.1, 0.5, and 1.0 mM). Cell lysis was performed by sonication, and soluble and insoluble protein fractions were separated by centrifugation. The expected molecular weight of the BOL-GST fusion protein (~60 kDa) is indicated by a red arrow. MM: molecular weight marker (PageRuler™ Prestained Protein Ladder, Thermo Scientific™, Vilnius, Lithuania). Gels were stained with Coomassie Brilliant Blue G-250. (**A**) SDS-PAGE (12%) gel of total protein extracts from cultures induced with IPTG concentrations of 0.05, 0.1, 0.5, and 1.0 mM. Lane 1: untransformed control. Lanes 2 and 7: non-induced control (total cell lysate); Lanes 3–6: total cell lysates induced with 0.05, 0.1, 0.5, and 1.0 mM IPTG, respectively; Lanes 8–9: 0.05 mM IPTG—soluble fractions (8), insoluble fraction (9); Lanes 10–11: 0.1 mM IPTG—soluble fractions (10), insoluble fraction (11); Lanes 12–13: 0.5 mM IPTG—soluble fractions (12), insoluble fraction (13); Lanes 14–15: 1.0 mM IPTG—soluble fractions (14), insoluble fraction (15). (**B**) Western blot using anti-GST antibody on whole-cell extracts from *E. coli* ArcticExpress (DE3) induced 24 h at 12 °C with 0.1 mM IPTG. The blot showed nonspecific signal patterns, hindering clear identification of bands corresponding to the recombinant protein. Lane 1: non-induced control (total cell lysate); Lane 2: total cell lysate post-induction at 0.1 mM IPTG; Lane 3: soluble fraction; Lane 4: insoluble fraction.

**Figure 5 ijms-26-08287-f005:**
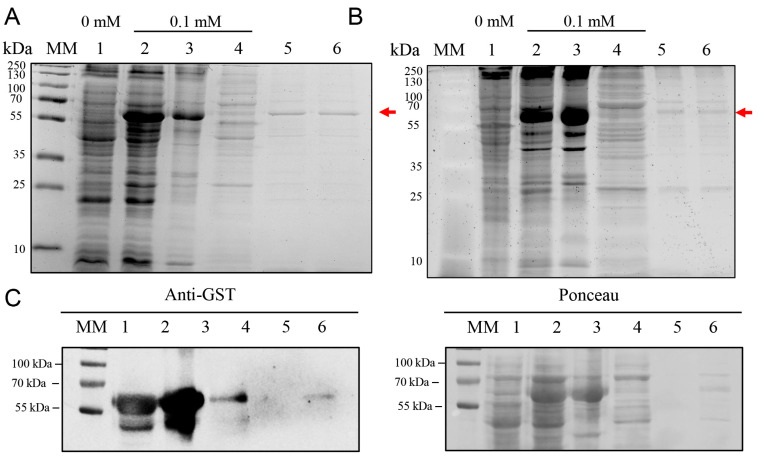
SDS-PAGE and Western blot analysis of BOL-GST expression in *E. coli* BL21(DE3)-R3-pRARE2 using the pGEX-4T-1 vector. Cultures were grown at 37 °C for 3 h and induced with 0.1 mM IPTG. The expected molecular weight of the fusion protein is indicated by a red arrow. MM: molecular weight marker (PageRuler™ Prestained Protein Ladder, Thermo Scientific™, Vilnius, Lithuania). (**A**) SDS-PAGE (12%) gel of samples from cultures induced at 37 °C for 4 h. Lane 1: non-induced control (total cell lysate); Lanes 2–4: total cell lysate induced with 0.1 mM IPTG (2), insoluble protein fraction (3), soluble protein fraction (4); Lanes 5–6: first and second elutions of BOL-GST purified from the soluble fraction using a Pierce™ GST Spin Purification column. Gel was stained with Coomassie Blue G-250. (**B**) SDS-PAGE (12%) gel of protein expression of cultures induced at 20 °C overnight. Lane 1: non-induced control (total cell lysate); Lanes 2–4: total cell lysate induced with 0.1 mM IPTG (2), insoluble protein fraction (3), soluble protein fraction (4); Lanes 5–6: first and second elutions of BOL-GST purified from the soluble fraction. Gel was stained with EZ-Blue. (**C**) Western blot using anti-GST antibody against cellular extracts from *E. coli* BL21(DE3)-R3-pRARE2 induced at 20 °C. Lane 1: non-induced fraction; Lane 2: total induced fraction; Lane 3: insoluble fraction; Lane 4: soluble fraction; Lane 5: no sample loaded (left intentionally blank for separation); Lane 6: purified BOL-GST protein. The antibody detected the presence of the fusion protein in both soluble and insoluble fractions, as well as in the purified sample.

**Figure 6 ijms-26-08287-f006:**
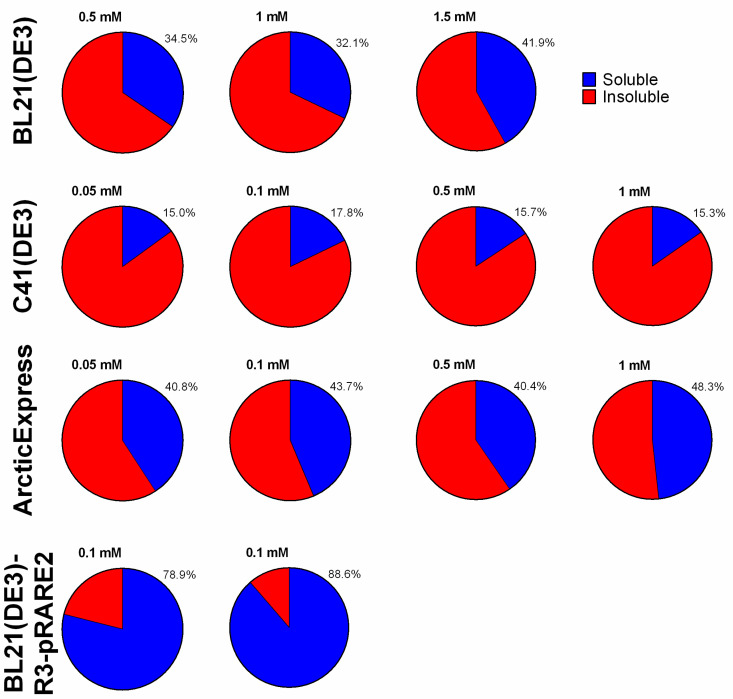
Semi-quantitative densitometric analysis of BOL distribution between soluble and insoluble fractions. Integrated densities of the BOL band were measured with ImageJ and normalized to the total integrated density of each lane (see Materials and Methods). Values are expressed as percentage of total lane signal (blue: soluble fraction; red: insoluble fraction). Raw images and the complete dataset are provided in Appendix A.

**Figure 7 ijms-26-08287-f007:**
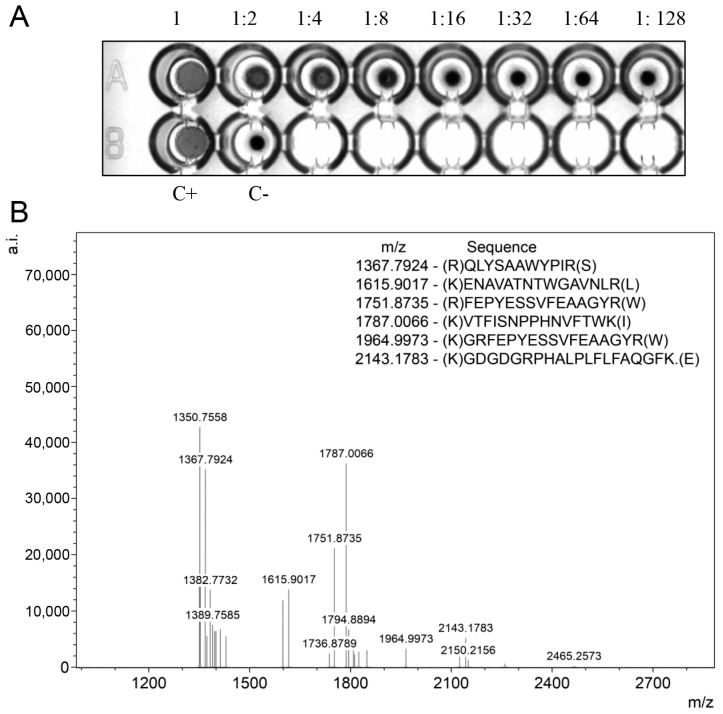
Functional activity and identity confirmation of recombinant BOL-GST. (**A**) Hemagglutination assay using 2% goat erythrocytes shows that the purified recombinant BOL-GST retains lectin activity, inducing visible agglutination observed up to the 1:8 dilution. In the bottom row, the first well contains plant-derived BOL lectin (positive control), the second well contains the saline solution (negative control), and the remaining wells were left empty. The image was acquired using the ChemiDoc Imaging System in grayscale to minimize lens distortion. (**B**) Mass spectrometric analysis of recombinant BOL-GST. Mass fingerprint of tryptic peptides was obtained using MALDI-TOF/TOF in positive ion mode, scanning from 640 to 3240 amu. Each tryptic peptide was subjected to LIFT fragmentation to generate a characteristic ion pattern, from which the amino acid sequences were deduced. Identified peptides matched the *Brassica oleracea* BOL lectin sequence in the NCBI database (MN937259), confirming protein identity.

## Data Availability

Data is contained within the article and Appendix A.

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
