# Peer review of "Recombinant Production of a TRAF-Domain Lectin from Cauliflower: A Soluble Expression Strategy for Functional Protein Recovery in E. coli"

_ijms, 2025, doi:10.3390/ijms26178287_

Round 1

Reviewer 1 Report

Comments and Suggestions for Authors

The submitted manuscript is very interesting in its subject matter. The work is methodological, showing the difficulties in obtaining recombinant plant lectins in bacterial expression systems and the possibilities of overcoming them. The possibility of using different types of vectors and E. coli strains to obtain recombinant cauliflower lectin is analyzed.

However, the article cannot be published in the presented form, since many questions about the illustrative material appear after reviewing the material, which must be resolved.

One question is about the methodological part of the work, which is not explained in the submitted manuscript. After reviewing the article listed in the list of references under No. 10 (2 co-authors of the submitted manuscript are among the co-authors of the article), it turned out that an attempt had already been made to obtain a recombinant protein in E. coli for this protein, however, in the submitted manuscript the conditions were optimized and the final yield of the recombinant protein was higher. But it is puzzling why, without additional explanation, section 2.1 provides data on an unsuccessful attempt to obtain a recombinant protein with His-tag (it seems necessary to make a reference and discussion with the previous article by the co-authors of the manuscript).

In addition, given the above, the phrase in line 224 about the limitation of the T7-based expression system is incorrect, since the pET21b plasmid was used in article released in 2020. It seems necessary to discuss in more detail why the use of this vector turned out to be more effective for BOL-6xHis than pET28a.

Despite the fact that line 80 states the intention to test various fusion tags, there is no discussion and explanation why the His-tag was not used when using the pGEX-4T-1 vector, and further stages of protocol optimization were carried out only using the GST-tag (which greatly changes the molecular weight of the recombinant protein).

In the legends to Fig. 1 and 2 the difference between lines 2 and 5, 3 and 8, 4 and 11 is not clear: what is the difference between “cultures induced with…” and “total induced fraction”. The legend needs to be clarified.

In the legend to Figure 3, the letter B does not indicate a Western blot.

Stained membranes are shown under the letter C (here again, there is an error in the caption to the Figure). However, the blot with anti-GST is not of the quality that is acceptable for publication. Another illustration of the membrane after visualization with antibodies should be provided.

In the legend to Figure 4, there is again a lot of confusion, so it was not possible to evaluate the presented results. A careful revision of the legend  is needed.

In the legend to Figure 5, there are also many inconsistencies. The Western blot does not have the six lanes designated, only 5, so it is difficult to evaluate the result described in the manuscript. A careful revision of the legend is needed.

Figure 6 in black and white does not allow to evaluate agglutination in samples 1:8 and 1:16 (as indicated in the legend). It is recommended to make a color illustration. In the bottom row in Figure A, the negative control is not in the last well, as indicated in the legend, but in the second.

The discussion does not compare the obtained data with the data from the article under No. 10 from the list of references on obtaining BOL-xHis recombinant protein.

The numerical values of the calculation described in section 4.9 (lines 401-402) are not given in the text of the manuscript.

In addition, it is necessary to trace the italicization of Latin names of biological objects in the text of the manuscript.

Reviewer 2 Report

Comments and Suggestions for Authors

The paper describes optimization of recombinant production of BOL lectin in several E. coli strains. While the paper has potential, there are several important issues that need to be addressed. 

1) Figures are of poor quality and blurry. The authors should improve their resolution since bands on gels can not be detected easily. 

2) Figure Legends are the main problem of the manuscript. The authors should carefully check and add missing information for Figures. Each band from gel or blot needs to be explained, otherwise results and conclusions are not clear.

3) Figure 3C, the authors state that this is SDS-PAGE gel. If so, the resolution is extremely bad and like this it can not contribute to the paper. Either improve the resolution or remove it completely. 

4) In the manuscript text it is stated that both Figures 5A and B represent expression at 20 °C while Figure 5 legend states otherwise. Please correct. Also, it seems that the expression is not so much dependent on the temperature, based on Figure 5A and B. However, purification of protein expressed at 37 °C resulted in much more purified protein, Figure 5A. Please add discussion about this. 

5) Please determine the yield of expression. Statement from line 181 is not clear enough. Compare the yield of soluble fractions from all of the tested conditions and strains. 

6) Hemagglutination assay is not clear. Do positive control (C+) and first sample from upper part (1) have the same lectin concentration since their agglutination is very different, with the one from positive control having better activity. Please discuss this. The authors should test the same concentrations of lectins and compare them. Also, activity in samples disappears from 1:4. In general this assay failed to give reliable results and this could be due to concentration or the fact that lectin has GST as well, which hinders its activity.  

Reviewer 3 Report

Comments and Suggestions for Authors

Introduction

  1. " These interactions enable lectins to agglutinate erythrocytes and precipitate glycoconjugates [1]" Line 36-37. Please clarify this statement. What interactions exactly? What with what? I also suggest describing this point below, in the paragraph with lines 42-45.
  2. BOL - must be deciphered!

  1. TRAF - decipher. What carbohydrates exactly does this domain bind?
  2. What limitations of plant extraction prevent the isolation of lectin from cabbage?
  3. MATH - decipher. I suggest first characterizing this domain, and then telling about your successes. Currently, lines 69-76 are written very confusingly.

Results

  1. IPTG - decipher!
  2. What is the molecular weight of the BOL-GST fuse

Discussion.

  1. «In the present study, no detectable expression of BOL-6xHis was observed, which may be attributed to several intrinsic limitations of the T7-based expression system». lines 223-224. Is T7 a promoter? It is necessary to first indicate that it is a promoter, and only then explain in the next sentence what the problem is. Please correct the text. Where is this promoter located?
  2. " In contrast to BOL-6xHis, the efficiency of the BL21(DE3) strain has been demon- 232 strated in other contexts ". Line 232-233. Are you comparing the BOL-6xHis protein and the BL21(DE3) strain? Correct the text.
  3. Why was the pET28a vector and the BL21(DE3) and C41(DE3) bacterial strains initially used? Justify. Perhaps this vector would have been effective with other strains? 11. Why did you use the pGEX-4T-1 vector and the BL21(DE3) strain? How do the vectors differ?
  4. Explain why BOL-GST expression was more efficient? What were the prerequisites for this experiment?

Round 2

Reviewer 1 Report

Comments and Suggestions for Authors

The comments were taken into account, changes were made to the manuscript, the quality of the illustrative material was improved.

However, a question arose regarding the newly introduced fig. 6, densitometry data.

Such quantitative analysis is correct to carry out only in the case of the equal protein loading. The illustrations of SDS-analysis given in the manuscript are not balanced for protein loading. Different amounts of protein were added to the gel wells, which is visible from the intensity. In the case of a qualitative comparison, to answer the question of whether the protein of interest are presented in the sample or not, disregard the alignment of the protein loading to the gel wells can be considered acceptable.

In the case of quantitative calculations, careful control of the protein loading and the introduction of an internal standard are necessary, the intensity of the band of which will be recalculated for the results.

In the presented version, it is incorrect to provide densitometry data.

Author Response

Comments 1: The comments were taken into account, changes were made to the manuscript, the quality of the illustrative material was improved. 

However, a question arose regarding the newly introduced fig. 6, densitometry data.

Such quantitative analysis is correct to carry out only in the case of the equal protein loading. The illustrations of SDS-analysis given in the manuscript are not balanced for protein loading. Different amounts of protein were added to the gel wells, which is visible from the intensity. In the case of a qualitative comparison, to answer the question of whether the protein of interest are presented in the sample or not, disregard the alignment of the protein loading to the gel wells can be considered acceptable.

In the case of quantitative calculations, careful control of the protein loading and the introduction of an internal standard are necessary, the intensity of the band of which will be recalculated for the results.

In the presented version, it is incorrect to provide densitometry data.

Response 1: Thank you for your thoughtful comment. We agree that quantitative densitometry ideally requires equal protein loading or the use of an internal loading control. As re-running the gels was not feasible, we addressed this point by re-analyzing the original gel images using ImageJ. For each lane, the integrated density of the BOL band was background-subtracted and expressed as a percentage of the total integrated density of the same lane, here referred to as % band. 
From these normalized values, we calculated: (i) % total soluble protein in the production, (ii) % total insoluble protein, and (iii) % soluble protein (fraction of BOL present in the soluble fraction relative to total BOL in production). These values are semi-quantitative and reflect relative protein distribution between fractions, rather than absolute amounts. To ensure clarity, we have updated the Figure legend (lines 245–249, p. 9) and expanded the Methods section (lines 438–446, p. 14) to describe the normalization and calculation steps explicitly, and to state that the results are semi-quantitative.In addition, the raw gel images and the data tables used for this analysis are now available in the Supplementary Information for transparency and reference.

Sincerely,

The authors

Reviewer 2 Report

Comments and Suggestions for Authors

I thank the authors for addressing all of my issues. The quality of the paper and figures is improved a lot.  

Author Response

Comments 1: I thank the authors for addressing all of my issues. The quality of the paper and figures is improved a lot.  

Response 1: We sincerely thank the reviewer for the positive feedback and for the valuable suggestions provided during the review process. We are pleased to know that the revisions have improved the quality of the paper and figures. Your comments and recommendations greatly contributed to strengthening our work.

Sincerely,

The authors